# ResGrad: Residual Denoising Diffusion Probabilistic Models for Text to Speech

## Abstract

Denoising Diffusion Probabilistic Models (DDPMs) are emerging in text-to-speech (TTS) synthesis because of their strong capability of generating high-fidelity samples. However, their iterative refinement process in high-dimensional data space results in slow inference speed, which restricts their application in real-time systems. Previous works have explored speeding up by minimizing the number of inference steps but at the cost of sample quality. In this work, to improve the inference speed for DDPM-based TTS model while achieving high sample quality, we propose ResGrad, a lightweight diffusion model which learns to refine the output spectrogram of an existing TTS model (e.g., FastSpeech 2) by predicting the residual between the model output and the corresponding ground-truth speech. ResGrad has several advantages: 1) Compare with other acceleration methods for DDPM which need to synthesize speech from scratch, ResGrad reduces the complexity of task by changing the generation target from ground-truth mel-spectrogram to the residual, resulting into a more lightweight model and thus a smaller real-time factor. 2) ResGrad is employed in the inference process of the existing TTS model in a plug-and-play way, without re-training this model. We verify ResGrad on the single-speaker dataset LJSpeech and two more challenging datasets with multiple speakers (LibriTTS) and high sampling rate (VCTK). Experimental results show that in comparison with other speed-up methods of DDPMs: 1) ResGrad achieves better sample quality with the same inference speed measured by real-time factor; 2) with similar speech quality, ResGrad synthesizes speech faster than baseline methods by more than 10 times. Audio samples are available at https://resgrad1.github.io/.

## 1 Introduction

In recent years, text-to-speech (TTS) synthesis (Tan et al., 2021) has witnessed great progress with the development of deep generative models, e.g., auto-regressive models (Oord et al., 2016; Mehri et al., 2017; Kalchbrenner et al., 2018), flow-based models (Rezende & Mohamed, 2015; van den Oord et al., 2018; Kingma & Dhariwal, 2018), variational autoencoders (Peng et al., 2020), generative adversarial networks (Kumar et al., 2019; Kong et al., 2020; Binkowski et al., 2020), and denoising diffusion probabilistic models (DDPMs, diffusion models for short) (Ho et al., 2020; Song et al., 2021b). Based on the mechanism of iterative refinement, DDPMs have been able to achieve a sample quality that matches or even surpasses the state-of-the-art methods in acoustic models (Popov et al., 2021; Huang et al., 2022b), vocoders (Chen et al., 2021; Kong et al., 2021b; Lee et al., 2022a; Chen et al., 2022; Lam et al., 2022), and end-to-end TTS systems (Huang et al., 2022a).

A major disadvantage of diffusion models is the slow inference speed which requires a large number of sampling steps (e.g., $100 \sim 1000$ steps) in high-dimensional data space. To address this issue, many works have explored to minimize the number of inference steps (e.g., $2 \sim 6$ steps) for TTS (Huang et al., 2022b; Liu et al., 2022c) or vocoder (Chen et al., 2021; Kong et al., 2021b; Lam et al., 2022; Chen et al., 2022). However, the high-dimensional data space cannot be accurately estimated with a few inference steps; otherwise, the sample quality would be degraded.

Instead of following previous methods to minimize the number of inference steps in DDPMs for speedup, we propose to reduce the complexity of data space for DDPM to model. In this way, the

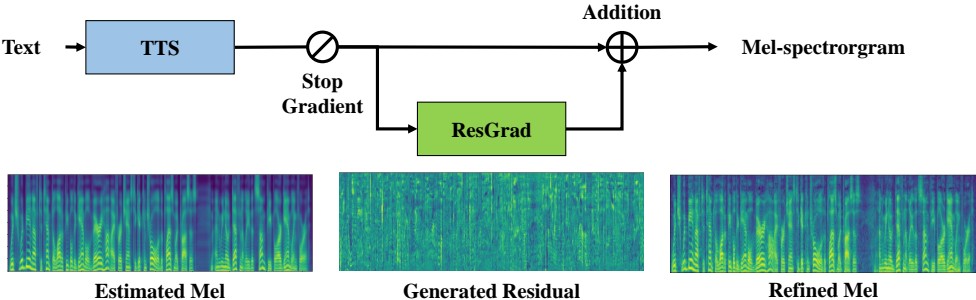

**Figure 1:** Illustration of ResGrad. ResGrad first predicts the residual between the mel-spectrogram estimated by an existing TTS model and the ground-truth mel-spectrogram, and then adds the residual to the estimated mel-spectrogram to get the refined mel-spectrogram.

model size as well as the number of inference steps can be reduced naturally, resulting in small real-time factor during inference. To the end, we propose ResGrad, a diffusion model that predicts the residual between the output of an existing TTS model and the corresponding ground-truth mel-spectrogram. Compared with synthesizing speech from scratch, predicting the residual is less complex and easier for DDPM to model. Specifically, as shown in Figure 1, we first utilize an existing TTS model such as FastSpeech 2 (Ren et al., 2021) to generate speech. In training, ResGrad is trained to generate the residual, while in inference, the output of ResGrad is added to the original TTS output, resulting in a speech with higher quality.

ResGrad can generate high quality speech with a lightweight model and small real-time factor due to the less complex learning space, i.e., residual space. Meanwhile, ResGrad is a plug-and-play model, which does not require retraining the existing TTS models and can be easily applied to improve the quality of any existing TTS systems.

The main contributions of our work are summarized as follows:

- We propose a novel method, ResGrad, to speed up the inference for DDPM model in TTS synthesis. By generating the residual instead of generating whole speech from scratch, the complexity of data space is reduced, which enables ResGrad to be lightweight and effective for generating high quality speech with a small real-time factor.
- The experiments on LJ-Speech, LibriTTS, and VCTK datasets show that compared with other speed-up baselines for DDPMs, ResGrad can achieve higher sample quality than baselines with the same RTF, while being more than 10 times faster than baselines when generating speech with similar speech quality, which verifies the effectiveness of ResGrad.

## 2 RELATED WORK

Denoising diffusion probabilistic models (DDPMs) have achieved the state-of-the-art generation results in various tasks (Ho et al., 2020; Song et al., 2021b; Kingma et al., 2021; Dhariwal & Nichol, 2021; Ramesh et al., 2022) but require a large number of inference steps to achieve high sample quality. Various methods have been proposed to accelerate the sampling process (Song et al., 2021a; Vahdat et al., 2021; Dhariwal & Nichol, 2021; Zheng et al., 2022; Xiao et al., 2022; Salimans & Ho, 2022).

In the field of speech synthesis, previous works on improving the inference speed of DDPMs can be mainly divided into three categories: 1) The prior/noise distribution used in DDPMs can be improved by leveraging some condition information (Lee et al., 2022a; Popov et al., 2021; Koizumi et al., 2022). Then, the number of sampling steps in inference stage can be reduced by changing the prior/noise distribution from standard Gaussian to reparameterized distribution. 2) The inference process can start from a latent representation which is acquired by adding noise on the additional input, instead of starting from standard Gaussian noise, which can reduce the number of inference steps in sampling process (Liu et al., 2022b). 3) Additional training techniques can be used for DDPMs. For example, DiffGAN-TTS (Liu et al., 2022c) utilizes GAN-based models in the framework of DDPMs to replace multi-step sampling process with the generator.

The above methods improve the inference speed of DDPMs from different perspectives. However, to guarantee the high sample quality of generated speech, sufficient number of sampling steps is required by these methods. Otherwise, the speech quality would be largely degraded.

In this work, instead of generating the whole speech from scratch, we generate the residual between the output of an existing TTS model and the corresponding ground-truth speech. The residual space is much easier than the origin data space for DDPMs to model, and thus a lightweight DDPM model can be trained and inferred with small sampling steps, resulting in a small real-time factor. Note that as we improve the sample quality upon existing TTS models (by predicting the residual), a small number of inference steps are enough to enhance the sample quality of estimated mel-spectrogram from an existing TTS model, and we can achieve the comparable sample quality which is on par with the quality achieved by a larger DDPM model predicting from scratch with much more sampling steps.

Beyond speech synthesis, there are some relevant work to speed up the inference of DDPMs. In Whang et al. (2022), they jointly trained a regression-based model for initial image deblurring and a diffusion model for the residual estimation, which effectively improved the image quality measured by objective distances. We separately train ResGrad to generate residual information, which does not require jointly training with other models and carefully developing new training configurations.

## 3 METHOD

In this section, we first give an overview of proposed ResGrad, and then introduce the formulation of denoising diffusion probabilistic models in ResGrad. At last, we discuss the advantages our method.

### 3.1 OVERVIEW OF RESGRAD

As shown in Figure 1, ResGrad is trained to predict the residual between the mel-spectrogram estimated by an existing TTS model and the ground-truth mel-spectrogram, and then adds the residual to the estimated mel-spectrogram to get the refined mel-spectrogram. The generated residual contains high-frequency details and avoid over-smoothness in the estimated mel-spectrogram, and thus can improve the sample quality. The three mel-spectrograms in Figure 1 show the estimated mel-spectrogram from an existing TTS model (i.e., FastSpeech 2), the residual generated by ResGrad, and the refined mel-spectrogram that is summation of the first two.

### 3.2 FORMULATION OF RESGRAD

We introduce the formulation of ResGrad, which leverages a denoising diffusion probabilistic model to learn the residual between the estimated and ground-truth mel-spectrograms. The residual $x$ is calculated as

$$x = mel_{GT} - f_\psi(y),\tag{1}$$

where $mel_{GT}$ denotes the ground-truth mel-spectrogram, $f_\psi$ represents an existing TTS model (e.g., FastSpeech 2 (Ren et al., 2021)), $y$ denotes the text input.

In the training stage, we injects standard Gaussian noise $\epsilon \sim \mathcal{N}(0, I)$ into the clean residual data $x_0$ according to a predefined noise schedule $\beta$ with $0 < \beta_1 < \cdots < \beta_T < 1$. At each time step $t \in [1, \ldots, T]$, the destroyed samples are calculated as:

$$q(x_t|x_0) = \mathcal{N}(x_t; \sqrt{\bar{\alpha}_t}x_0, (1 - \bar{\alpha}_t)\epsilon).\tag{2}$$

where the $\alpha_t := 1 - \beta_t$, and $\bar{\alpha}_t := \prod_{s=1}^{t} \alpha_s$ denotes a corresponding noise level at time step $t$.

There are different methods to parameterize DDPMs and derive the training objective (Kingma et al., 2021). We follow the previous work Grad-TTS (Popov et al., 2021) to directly estimate the score function $s_\theta$ which means the gradient of the noised data log-density $\log q(x_t|x_0)$ with respect to the data point $x_t$: $s(x_t, t) = \nabla_{x_t} \log q(x_t|x_0) = -\frac{\epsilon}{\sqrt{1-\bar{\alpha}_t}}$. Our training objective is then formulated as:

$$L(\theta) = \mathbb{E}_{x_0,\epsilon,t} \left\| s_\theta(x_t, t, c) + \frac{\epsilon}{\sqrt{1 - \bar{\alpha}_t}} \right\|_2^2.\tag{3}$$

In the inference stage, we first estimate the mel-spectrogram $f_\psi(y)$ from the text input $y$ with an existing TTS model $f_\psi$. By conditioning on $f_\psi(y)$, we start from standard Gaussian noise $p(x_T) \sim \mathcal{N}(0, I)$, and iteratively denoise the data sample to synthesize the residual $x_0$ by:

$$p_\theta(x_0, \cdots, x_{T-1}|x_T, c) = \prod_{t=1}^{T} p_\theta(x_{t-1}|x_t, c), \qquad (4)$$

where $c$ is the generated mel-spectrogram $f_\psi(y)$ from an existing TTS model. Then, we add the generated residual $\hat{x}_0$ and $f_\psi(y)$ together as the final output

$$mel_{ref} = \hat{x}_0 + f_\psi(y), \qquad (5)$$

where $mel_{ref}$ is the refined mel-spectrogram.

### 3.3 Advantages of ResGrad

Generally speaking, both iterative TTS models (e.g., GradTTS) and non-iterative TTS models (e.g., FastSpeech 2) have their pros and cons: 1) iterative TTS models can achieve high sample quality, but at the cost of slow inference speed; 2) non-iterative TTS models are fast in inference, but the sample quality may be not good enough. Instead of choosing either iterative or non-iterative method, ResGrad makes full use of them: using a non-iterative model to predict a rough sample, and using an iterative model to predict the residual. We analyze the advantages of ResGrad from the following two aspects:

- Standing on the shoulders of giants. ResGrad fully leverages the prediction of an existing non-iterative TTS model and just predicts the residual in an iterative way. Compared to previous methods on speeding up diffusion models that predict the data totally from scratch, ResGrad stands on the shoulders of giants and avoids reinventing wheels, which leaves large room for speeding up diffusion models.
- Plug-and-play. ResGrad does not requires re-training the existing TTS models but just acts like a post-processing module, which can be applied on existing TTS models in a plug-and-play way.

## 4 Experimental Setup

### 4.1 Dataset and Preprocessing

**Dataset** We conducted our experiments on three open-source benchmark datasets. **LJ-Speech** (Ito & Johnson, 2017) [1] is a single-speaker dataset with a sampling rate of 22.05kHz. It contains $13,100$ English speech recordings from a female speaker, with around 24 hours. Following Kong et al. (2021b); Chen et al. (2022), we use the 523 samples in LJ-001 and LJ-002 samples for model evaluation, the remaining $12,577$ samples for model training. **LibriTTS** (Zen et al., 2019) [2] is a multi-speaker dataset with a sampling rate of 24kHz. In total, we use 1046 speakers from the clean subset (train-clean-100, train-clean-360, dev-clean, test-clean). For each speaker, we randomly select 20 recordings as the test dataset, while the remaining samples are used as training dataset. **VCTK** (Yamagishi, 2012) [3] is a multi-speaker dataset with a sampling rate of 48kHz. We extract 108 speakers from the datasets, and the first 5 recordings of each speaker (540 recordings in total) are used as the test dataset. The remaining samples are used for training.

**Preprocessing** We use the open-source tools (Park, 2019) to convert the English grapheme sequence to phoneme sequence, and then transform the raw waveform of above three datasets into mel-spectrogram following the common practice. Specifically, for LJSpeech, we extract 80-band mel-spectrogram with the FFT 1024 points, 80Hz and 7600Hz lower and higher frequency cutoffs, and a hop length of 256 (Ren et al., 2021; Huang et al., 2022b; Popov et al., 2021). For LibriTTS, we extract 80-band mel-spectrogram with the FFT 2048 points, 80Hz and 7600Hz lower and higher frequency cutoffs, and a hop length of 300 (Lee et al., 2022b; Kim et al., 2020). For VCTK, we use 128-band mel-spectrogram with the FFT 2048 points, 0Hz and 24000Hz lower and higher frequency cutoffs, and a hop length of 480 (Jang et al., 2021; Liu et al., 2022a).

---

[1] https://keithito.com/LJ-Speech-Dataset/
[2] https://research.google/tools/datasets/LibriTTS/
[3] https://datashare.ed.ac.uk/handle/10283/3443

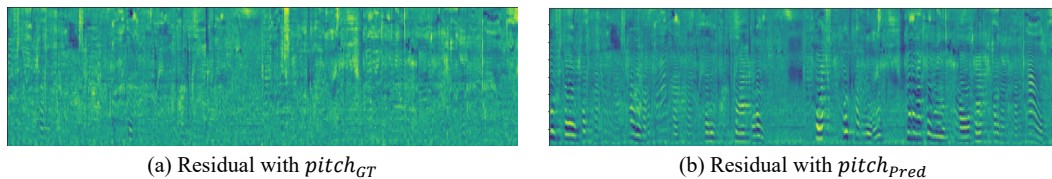

(a) Residual with $pitch_{GT}$              (b) Residual with $pitch_{Pred}$

**Figure 2:** The comparison between the residual calculated with ground-truth pitch, and the residual calculated without ground-truth pitch (i.e., pitch predicted by model).

**Calculation of Residual** We calculate the residual between the mel-spectrogram predicted by FastSpeech 2 (Ren et al., 2021) and the ground-truth mel-spectrogram. As FastSpeech 2 utilizes ground-truth pitch/duration in training while predicted pitch/duration in inference, we need to consider whether use the ground-truth or predicted pitch/duration to calculate the residual. For duration, we should use ground-truth duration to predict mel-spectrogram for residual calculation. Otherwise there will be length mismatch between the predicted and ground-truth mel-spectrograms. For pitch, we compare the residual calculated by using ground-truth and predicted pitch, as shown in Figure 2(a) and Figure 2(b). In Figure 2(a), the residual represents more about the high-frequency details in ground-truth mel-spectrogram that are difficult to be predicted by TTS models. In this way, the residual generated ResGrad mainly adds the high-frequency details that can improve the samples quality, e.g., the clearness and the naturalness, instead of changing the pitch of estimated mel-specotogram. However, if we calculate the residual with predicted pitch as shown in Figure 2(b), ResGrad will simultaneously correct the pitch in estimated samples towards the ground-truth samples, which will increase the burden of ResGrad. Thus, we use ground-truth pitch to calculate the residual. Note that in inference, the estimated mel-spectrogram is predicted by FastSpeech 2 with both predicted duration and pitch, instead of ground-truth pitch and duration.

## 4.2 MODEL CONFIGURATION

We choose FastSpeech 2 (Ren et al., 2021) as the non-iterative based TTS model, which is composed of 6 feed-forward Transformer blocks with multi-head self-attention and 1D convolution on both phoneme encoder and mel-spectrogram decoder. In each feed-forward Transformer, the hidden size of multi-head attention is set to 256 and the number of head is set to 2. The kernel size of 1D convolution in the two-layer convolution network is set to 9 and 1, and the input/output size of the number of channels in the first and the second layer is 256/1024 and 1024/256. As for the duration predictor and variance adaptor, the architecture we use is exactly the same as that in FastSpeech2 (Ren et al., 2021), which are composed of stacks of several convolution networks and the final linear projection layer. The convolution layers of the duration predictor and variance adaptor are set to 2 and 5, the kernel size is set to 3, the input/output size of all layers is 256/256, and the dropout rate is set to 0.5. ResGrad is a U-Net architecture, which mainly borrows the implementation of the official open-source in Grad-TTS (Popov et al., 2021), except that we have smaller model parameters.

For LJ-Speech and LibriTTS, we use the publicly available pretrained HiFi-GAN (Kong et al., 2020)[4]. For VCTK, we trained the HiFi-GAN to generate 48kHz waveform with the open-source code[5]. In experiments, the same vocoder is used for each of the baseline TTS models.

## 4.3 TRAINING AND EVALUATION

For the training of non-iterative based TTS model, we use the AdamW optimizer with $\beta_1 = 0.8$, $\beta_2 = 0.99$, weight delay $\lambda = 0.01$, and follow the same learning rate plan schedule in Vaswani et al. (2017). We use 8 NVIDIA V100 GPUs, the batch size is set to 32 per GPU. It takes 100k steps until moderate quality speech can be generated.

For the training of ResGrad, the Adam optimizer is used and the learning rate is set to 0.0001, The number of diffusion time steps is set as T = 1000. We use 8 NVIDIA V100 GPUs, the batch size is

---

[4]https://github.com/kan-bayashi/ParallelWaveGAN
[5]https://github.com/jik876/hifi-gan

**Table 1:** The MOS and RTF results of different methods on LJSpeech, LibriTTS, and VCTK datasets. Note that GradTTS-50, ResGrad-4, and ResGrad-50 mean using 50, 4, and 50 iteration steps respectively. The MOS scores on LibriTTS dataset is lower than LJSpeech and VCTK. There are two main reasons. Firstly, the speech samples in that dataset may contain background noise, which decreases the quality of synthesized samples. Secondly, LibriTTS contains 1046 speakers. As other models are generating TTS samples from scratch, the synthesis task is challenging. In comparison, generating the residual part of each speaker becomes easier thus achieving better sample quality.

| Models | LJSpeech | | LibriTTS | | VCTK | |
|---|---|---|---|---|---|---|
| | MOS ($\uparrow$) | RTF ($\downarrow$) | MOS ($\uparrow$) | RTF | MOS ($\uparrow$) | RTF |
| Recordings | $4.91 \pm 0.02$ | − | $4.4 \pm 0.05$ | − | $4.75 \pm 0.02$ | − |
| GT mel+ Vocoder | $4.28 \pm 0.06$ | − | $4.5 \pm 0.04$ | − | $4.61 \pm 0.04$ | − |
| FastSpeech 2 | $3.29 \pm 0.10$ | 0.003 | $2.58 \pm 0.13$ | 0.003 | $3.14 \pm 0.11$ | 0.004 |
| DiffGAN-TTS | $2.14 \pm 0.09$ | 0.009 | − | − | − | − |
| DiffSpeech | $3.78 \pm 0.11$ | 0.182 | $2.50 \pm 0.13$ | 0.263 | $3.44 \pm 0.10$ | 0.857 |
| ProDiff | $3.14 \pm 0.09$ | 0.033 | $2.75 \pm 0.11$ | 0.061 | $3.29 \pm 0.09$ | 0.235 |
| GradTTS-50 | $\mathbf{4.14 \pm 0.09}$ | 0.193 | $2.38 \pm 0.18$ | 0.259 | $3.85 \pm 0.09$ | 0.292 |
| ResGrad-4 | $4.07 \pm 0.11$ | 0.018 | $3.00 \pm 0.16$ | 0.020 | $3.78 \pm 0.11$ | 0.022 |
| ResGrad-50 | $\mathbf{4.14 \pm 0.09}$ | 0.169 | $\mathbf{3.41 \pm 0.18}$ | 0.223 | $\mathbf{3.93 \pm 0.09}$ | 0.244 |

set to 16 per GPU. we restrict the max length of each sample as 4 seconds to enlarge the batch size. And we find that the strong sample quality improvement can be achieved after 500k training steps.

For evaluation, we mainly choose human subjective evaluation: **MOS (Mean Opinion Score)** and **CMOS (Comparison Mean Opinion Score)** tests. We invite 14 native English speakers as judges to give the MOS for each utterance to evaluate the overall sample quality with a 5-point scale. Each CMOS result is given by 14 judges per utterance for comparing the samples generated by the two different models. For measuring the inference speed, we use the objective measurement real-time factor (RTF) calculated on an NVIDIA V100 GPU device.

## 4.4 TRAINING OF BASELINE MODELS

For a fair and reproducible comparison, we train several baseline models with the same dataset, i.e. the single speaker dataset LJSpeech and the multi-speaker dataset VCTK and LibriTTS. The details of the training settings of these baseline models are: 1) DiffSpeech is an acoustic model for TTS based on diffusion model. We train DiffSpeech following Liu et al. (2022b), and the configurations are adjusted accordingly on different data set. For DiffSpeech, the training has two stages: warmup stage and main stage. Warmup stage trains the auxiliary decoder for 160k steps, which is FastSpeech 2 (Ren et al., 2021) here. Main stage trains DiffSpeech for 160k steps until convergence. 2) DiffGAN-TTS. Following Liu et al. (2022c), we train DiffGAN-TTS with T=1, 2, and 4. For both the generator and the discriminator, we use the Adam optimizer, with $\beta_1 = 0.5$ and $\beta_2 = 0.9$. Models are trained using one NVIDIA V100 GPU. We set the batch size as 64, and train models for 300k steps until loss converge. 3) ProDiff (Huang et al., 2022b) is an progressive fast diffsion model for high-quality text-to-speech. It is distilled from the N-step teacher model, which further decreases the sampling time. Following Huang et al. (2022b), we first train the ProDiff teacher with 4 diffusion steps. Then, we take the converged teacher model to train ProDiff with 2 diffusion steps. Both the ProDiff teacher model and ProDiff model are trained for 200,000 steps by using one NVIDIA V100 GPU. 4) GradTTS. Following Popov et al. (2021), we train model on original mel-spectrograms. Grad-TTS was trained for 1,700k steps on 8 NVIDIA V100GPU with a batch size of 16. We chose Adam optimizer and set the learning rate to 0.0001. 5) FastSpeech 2. Following Ren et al. (2021), we train FastSpeech 2 with a batch size of 48 sentences. FastSpeech 2 is trained for 160k steps until convergence by using 8 NVIDIA V100 GPU.

# 5 RESULTS

## 5.1 SPEECH QUALITY AND INFERENCE SPEED

We compare ResGrad with several baselines, including 1) Recording: the ground-truth recordings; 2) GT mel + Vocoder: using ground-truth mel-spectrogram to synthesize waveform with HiFi-GAN vocoder (Kong et al., 2020); 3) FastSpeech 2 (Ren et al., 2021): one of the most popular non-autoregressive TTS model which can synthesis high quality speech in fast speed, 4) DiffGAN-TTS (Liu et al., 2022c) [6], a DDPM-based text-to-speech model which speeds up inference with GANs, 5) DiffSpeech (Liu et al., 2022b) [7], a DDPM-based text-to-speech model which speed up inference by a shallow diffusion mechanism; 6) ProDiff (Huang et al., 2022b)[8], an progressive fast diffusion model for text-to-speech, 7) GradTTS (Popov et al., 2021)[9], a DDPM-based TTS model which can synthesis high-quality speech. FastSpeech 2 is the non-iterative TTS model used in this work. Other baselines provide different methods to accelerate the inference speed of DDPMs.

We maintain the consistency of experimental settings between different methods to exclude other interference factors, and re-produce the results of these methods on the three open-source datasets. The MOS test and RTF results are shown in Table 1 (ResGrad-4 and ResGrad-50 represent using 4 and 50 iteration steps respectively). We have several observations:

- Voice quality. Our ResGrad-50 achieves the best MOS in all the three datasets, especially in multi-speaker and high sampling rate datasets, i.e. LibriTTS and VCTK. In LJSpeech, ResGrad-50 outperforms all baselines, and achieves on-par MOS with GradTTS-50, but with a faster inference speed. Also, ResGrad-4 outperforms all other baselines except GradTTS-50 in LJSpeech and VCTK. In LibriTTS, ResGrad-4 even performs better than GradTTS-50.
- Inference speed. Our ResGrad-4 outperforms other baseline considering both inference speed and voice quality. Compare with GradTTS-50, both ResGrad-4 and ResGrad-50 achieves faster inference speed with on-par or even better speech quality. Compared with ProDiff and Diff-Speech, ResGrad-4 reduces the inference time, as well as achieving better voice quality in the three datasets. The inference speed of DiffGAN-TTS is the fastest, but the voice quality is very poor.

As a summary, the quality of our ResGrad with 4-step iteration is much higher than that of the non-iterative based TTS method (e.g., FastSpeech 2) and other speech-up method of DDPMs (e.g., DiffGAN-TTS, DiffSpeech and ProDiff). With the same iteration step, our ResGrad can match or even exceed the quality of Grad-TTS model, but enjoys a smaller model and lower RTF. It is worth mentioning that our advantages are more obvious in more challenging datasets with multiple speakers and high sampling rate.

The comments from judges can be found in Appendix A.

## 5.2 ABLATION STUDY

**Residual Calculation**   As described in Section 4.1, there are two choices for calculating the residual: using $pitch_{GT}$ or $pitch_{Pred}$ in FastSpeech 2. We show both of the calculated residual samples and the generated one in Figure 3. As it can be seen, when we use $pitch_{Pred}$, the residual related to pitch information is not generated as it is in the ground-truth residual samples. In inference stage, it means the model can not correct the pitch towards the ground-truth data samples. Moreover, it may produce distortions in synthesized samples. We show the quality comparison between these two settings in Table 2.

---

[6]https://github.com/keonlee9420/DiffGAN-TTS
[7]https://github.com/MoonInTheRiver/DiffSinger
[8]https://github.com/Rongjiehuang/ProDiff
[9]https://github.com/huawei-noah/Speech-Backbones/tree/main/Grad-TTS

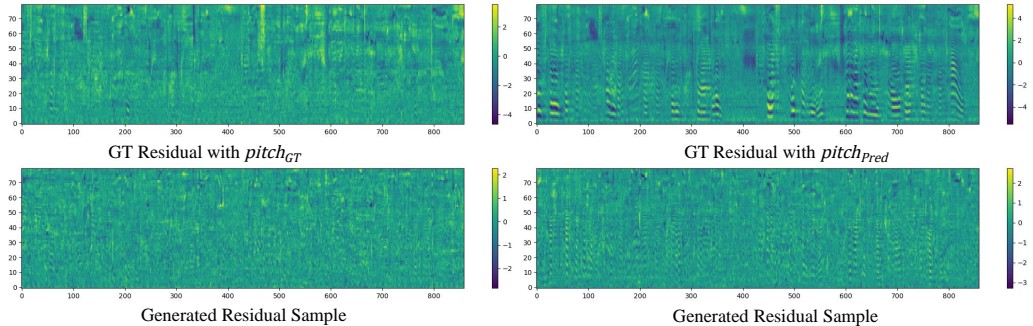

**Figure 3:** The left column shows a ground-truth residual sample calculated with both $dur_{GT}$ and $pitch_{GT}$ (top) and the corresponding predicted residual sample (bottom), while the right column shows the corresponding ground-truth residual sample calculated with $dur_{GT}$ and $pitch_{Pred}$ (top) and the corresponding predicted residual sample (bottom).

**Table 2:** The effect of GT pitch information.

| Model | CMOS (↑) |
|---|---|
| ResGrad | 0 |
| ResGrad $- pitch_{GT}$ | $-0.17$ |

**Table 3:** Comparison between ResGrad and ResUNet.

| Model | CMOS (↑) | Size (M) |
|---|---|---|
| ResGrad | 0.00 | 2.03 |
| ResUNet | $-0.18$ | 31.8 |

        (a) FastSpeech 2         (b) Ground Truth         (c) ResGrad

**Figure 4:** The comparison between the mel-spectrogram generated by FastSpeech 2, the refined mel-spectrogram by ResGrad, and the ground-truth mel-spectrogram. FastSpeech 2 suffers from the over-smoothing problem, while ResGrad generates mel-spectrogram with more detailed frequency information, which is more similar to ground truth mel-spectrogram.

**Residual Prediction**    It is natural to consider if other models can be used for predicting the residual information and effectively improving the sample quality. Following Liu et al. (2021), we use ResUNet (Kong et al., 2021a), which is an improved UNet (Ronneberger et al., 2015), to model the residual with the generated mel-spectrogram $f_\psi(y)$. The ResUNet consists of several encoder and decoder blocks, with skip connections between encoder and decoder blocks at the same level. Both encoder and decoder is a series of residual convolutions, and each convolutional layer in ResConv consists of a batch normalization, a leakyReLU activation, and a linear convolutional operation. Therefore, we employ the Residual UNet (ResUNet) network to predict the residual information for comparison. ResUNet has achieved promising results in predicting the high-frequency information of mel-spectrogram (Liu et al., 2022a) in speech super-resolution task (Wang & Wang, 2021). Compared with ResGrad, a considerably larger ResUNet model (31.8 M) is used for generating the residual. We show the comparison of sample quality, model size, and inference speed between Res-Grad and ResUNet in Table 3. It can be seen that ResGrad achieves higher sample quality but with much smaller model size, demonstrating the effectiveness of our method.

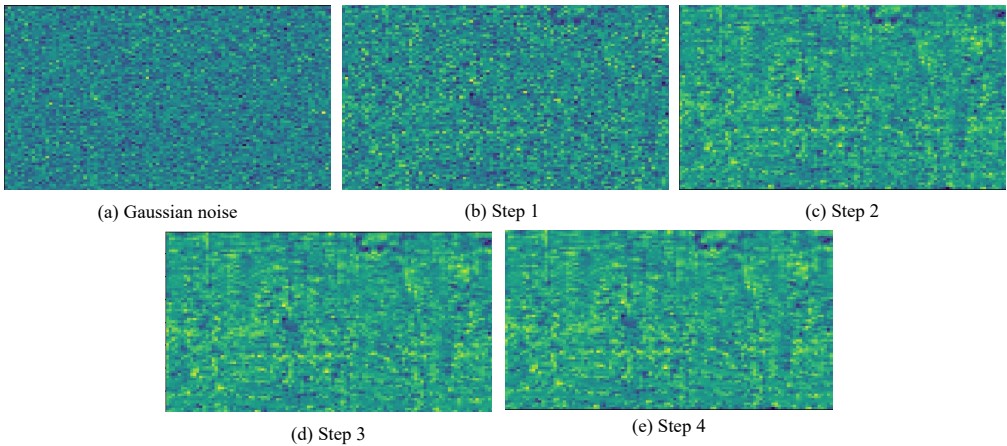

| (a) Gaussian noise | (b) Step 1 | (c) Step 2 |
|---|---|---|
| (d) Step 3 | (e) Step 4 | |

**Figure 5:** The decoding process of ResGrad in 4 sampling steps.

**Table 4:** CMOS results on LibriTTS.

| Model | CMOS (↑) | Size (M) | RTF (↓) |
|---|---|---|---|
| ResGrad | 0.00 | 2.03 | 0.223 |
| ResGrad-L | 0.09 | 7.68 | 0.265 |

**Table 5:** CMOS results on VCTK.

| Model | CMOS (↑) | RTF (↓) |
|---|---|---|
| ResGrad | 0 | 0.244 |
| ResGrad-L | 0.02 | 0.294 |

**Model Size** The performance of ResGrad on improving sample quality is not sensitive to the model size. We verify it by employing a large diffusion model (i.e., the decoder model in Grad-TTS (Popov et al., 2021)) in ResGrad for comparison. We represent it as ResGrad-L. The comparison results are shown in Table 4 and Table 5. With a same number of inference steps, 50, it achieves similar CMOS with ResGrad on both LibriTTS and VCTK. However, the inference speed becomes slower.

## 5.3 CASE STUDY

As shown in Figure 4, we conduct case study and visualize the mel-spectrograms from ground truth recording and generated by different TTS models, i.e. FastSpeech 2 and ResGrad. We can observe that FastSpeech 2 tend to generate mel-spectrograms which are over-smoothing. Compared with FastSpeech 2, ResGrad can generates mel-spectrograms with more detailed information. Therefore, ResGrad can synthesis more expressive and higher quality speech.

Figure 5 shows the inference process of ResGrad, which contains 4 sampling steps to generate the clean residual data from the standard Gaussian noise. As can be seen, the Gaussian noise are effectively removed in each sampling step. Although more details of residual samples will be generated with more inference steps, clean residual samples can be generated with a few sampling steps, which has been able to improve the TTS sample quality.

## 6 CONCLUSION

In this work, to accelerate the inference speed of diffusion-based TTS models, we propose ResGrad which models the residual between the output of an existing TTS model and the corresponding ground-truth. Different from all the previous works on speeding up diffusion models that learn the whole data space of speech, ResGrad learns the residual space that is much easier to model. As a result, ResGrad just requires a lightweight model with a few iterative steps to synthesize high-quality speech in a low real-time factor. Experiments on three datasets show that ResGrad is able to synthesize higher quality speech than baselines under same real-time factor and speed up baselines by more than 10 times under similar synthesized speech quality. For future work, we will investigate to apply the idea of ResGrad on other TTS model architectures or other tasks such as image synthesis.

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

## A  HUMAN EVALUATIONS AND COMMENTS

We list some comments about the perceptual quality of different models given by native English speakers. All of them are not familiar with TTS synthesis task.

**ResGrad-4**  "The model achieves high quality acoustics, limited articulation in speech, with very low (but audible) hiss in some recordings."; "pauses too long between the words".

**ResGrad-50**  "Words are really clear and understandable"; "slight less sharp compared to ground truth"; "much better acoustics - hiss level is reduced"; "Sounds robotic with buzzing noise present in some samples, although slower speech".

**DiffGAN-TTS**  "You can easily understand the words and distinguish them but the sound is quite distorted"; "Sounds distorted".

**DiffSpeech**  "low but noticeable hiss levels on most recordings, some recordings are buzzy. Pitch seems to be higher/speech is faster - is the sampling rate correct?"; "the speed make the quality lower and the volume drop sometimes".

**FastSpeech 2**  "similar comments to 1-1 (ResGrad-4)"; "similar to model 2 (DiffGAN-TTS), there is some good articulation, but very audible buzz that corrupts some of the speech."

**GradTTS-50**  "It make longer pauses when needed to better understand the meaning of the speech."; "articulation is very robotic. However, good acoustic quality (audible but quiet hiss in most recordings)"; "slower and more understandable. Realistic changes in pitch. More natural breaks between words which sound realistic.".

**ProDiff**  "Slight background noise but the human voice was mostly good"; "robotic voice and a bit faster of model 1-1 (ResGrad-4) making it a bit less understandable"; "limited articulation high level of crackle in most recordings".

**GT-Mel**  "articulation is good. High quality acoustics similar in 1-1 (ResGrad-4)"; "slower and more understandable. Lacks realistic changes in pitch. Slight high frequency artefact"; "very realistic audio with realistic tonal changes, some noise present in audio samples but I could listen to audio books with this voice".

