# OpenReview forum: "ResGrad: Residual Denoising Diffusion Probabilistic Models for Text to Speech"
_ICLR.cc/2023/Conference — Submitted to ICLR 2023_

### Official Review · Reviewer_DApv · 2022-10-23

**Confidence:** 5
**Correctness:** 3
**Technical Novelty And Significance:** 2
**Empirical Novelty And Significance:** 2
**Recommendation:** 3

**Clarity, Quality, Novelty And Reproducibility:**

The proposed method is clear but can be seen as the diffusion-based refinement model for FastSpeech 2. The method is similar to the existing model (Grad-TTS). Detailed explanations are shown in the weaknesses part.

**Strength And Weaknesses:**

Strengths
* Reduces the number of sampling steps of DDPM by modeling the residual.
* ResGrad shows comparable performance to the Grad-TTS and performs well with the 4 sampling steps.
* The demo page shows that ResGrad refines the sample quality of FastSpeech 2 well.

Weaknesses
* ***The proposed method is just the refinement model of the pre-trained FastSpeech 2.*** ResGrad can be applied to the specific deterministic TTS model (FastSpeech 2), but it seems difficult to apply to other TTS models. ResGrad requires the GT pitch and GT duration for training. It is challenging to model the residual between the output of other stochastic TTS models and GT mel.
* ***The proposed method is quite similar to Grad-TTS.*** The residual between the model output and GT mel can be modeled with DDPM in two ways. (1) Directly modeling the residual (GT mel - the model output). (2) Use the mean of the prior distribution with the model output and model the GT mel w/ DDPM. The latter method can be seen as the simple modification of Grad-TTS, which replaces the output of the text encoder with that of FastSpeech 2. There is no guarantee that the former method is more effective in reducing the sampling step than the latter method, which makes ResGrad similar to Grad-TTS.
* ***The baselines of the paper are the single-speaker TTS models.*** FastSpeech 2, DiffGAN-TTS, DiffSpeech, and Grad-TTS do not have experimental results for the multi-speaker TTS dataset (LibriTTS and VCTK) in their original papers. Therefore, these models may not perform well in multi-speaker settings by simply introducing the speaker embedding. For example, Grad-TTS shows very poor quality in the multi-speaker setting as can be seen from the demo page of this paper. To show that it works well on the multi-speaker dataset, it should compare with the multi-speaker TTS models.
* ***The generated samples from ResGrad sound a lot similar to the samples from FastSpeech 2 except for the sample quality.*** It is necessary to show whether the sample diversity of the diffusion-based generative model is lost by performing the residual modeling. It would be better to provide several samples of ResGrad-4 and ResGrad-50 for the same FastSpeech 2 output to the demo page.

**Summary Of The Paper:**

This paper proposes ResGrad, a diffusion-based model that models the residual between the model output and the GT mel. For efficient sampling, ResGrad refines the output of the pre-trained TTS model (FastSpeech 2) by modeling the residual. ResGrad speeds up the inference for DDPM by using the pre-trained TTS model. The experimental results show that ResGrad outperforms other baselines except for GradTTS and is comparable to GradTTS with the same sampling steps. For the single-speaker setup, the performance of ResGrad doesn’t deteriorate much with 4 sampling steps.

**Summary Of The Review:**

The method is limited to improving the output of FastSpeech 2, and it does not look much different from the existing model (Grad-TTS). The paper should compare ResGrad with the appropriate baselines for the multi-speaker datasets. The samples of ResGrad on the demo page have similar prosody (pitch & duration) to the samples of FastSpeech 2. It is required to show whether ResGrad does not lose sample diversity by modeling residuals. (which can be a possible disadvantage of the proposed methods)

---

### Official Review · Reviewer_U3kA · 2022-10-25

**Confidence:** 5
**Correctness:** 3
**Technical Novelty And Significance:** 2
**Empirical Novelty And Significance:** 2
**Recommendation:** 3

**Clarity, Quality, Novelty And Reproducibility:**

The idea of editing the outputs of a spectrogram predictor is not novel -- Tacotron 2 does this with its "post edit network", which is the first instance I have seen of it. Since Tacotron, GANs have been used to convert spectrograms from low-quality / over-smoothed to higher fidelity / crisper. This is the first application I have encountered of using diffusion models in this setting.

The manuscript is clear and easy to understand.

**Strength And Weaknesses:**

# Strengths

* Straightforward application of diffusion to improve a FastSpeech-based TTS system.

# Weaknesses
* Increases complexity in an unprincipled manner: The overall combination of FastSpeech and ResGrad is a mashup of a deterministic / non-probabilistic model and a deep generative model. Add in a vocoder to invert that spectrogram – say a HifiGAN or WaveNet and now you have yet another generative model in the stack. All of which were trained with different objectives and might interpret the spectrogram in different ways.
* Plug-and-play is not a strength, if the overall system is now a chain of 3 models (FastSpeech, ResGrad, and a vocoder). It would be more efficient to simply improve the vocoder to be able to work with oversmoothed spectrograms.
* The outputs of FastSpeech are oversmoothed and are likely a boring “average” prosody for the provided text due to the L1 or L2 loss used to predict prosodic details. Once it generates a spectrogram with prosodic decisions already made, you’ve lost access to the linguistic details that would be most informative for fixing the bad spectrogram it produced. ResGrad is not the best place to fix these decisions – the best place is in the FastSpeech model itself. If ResGrad changes the trajectory towards something that is more realistic or similar to the training data it may in doing so corrupt the prosodic meaning expressed in the original spectrogram (for example, if there were 5 valid ways to say something and the textual context strongly hinted at one of the 5 but the FastSpeech model was unable to realize it in a realistic way).  The ResGrad model has no way of knowing which of the valid ways of saying it it should go with, since it has no access to text or context that would resolve that question.
* Would like to see a CMOS comparison with GradTTS, which had equivalent MOS. MOS Is not a good tool for comparing models that have substantially similar performance.


**Summary Of The Paper:**

Predicts the residual between the output of a FastSpeech spectrogram predictor and the ground truth. Reduces the complexity of the problem the diffusion model is performing.


**Summary Of The Review:**

I appreciate the work that went into this and found it an interesting read.

The motivation of this work is to improve an existing deterministic spectrogram prediction system (FastSpeech 2) by improving the quality of its spectrogram outputs to appear more realistic. The flaw in this method is that the improvements have no way of resolving between multiple valid prosodic trajectories, since it has no access to the text.

The experimental results are not compelling – the method seems on-par with GradTTS-50, with a slight improvement in RTF. I suspect that if you pair FastSpeech 2 with a vocoder (e.g. a diffusion-based vocoder or WaveRNN, or HifiGAN) trained on its outputs, you could outperform or match the performance of ResGrad. It’s not clear why we would want to employ another, very complicated, deep generative model, when we already going to follow FastSpeech with a deep generative model like a HifiGAN – that could easily be trained to handle the over-smoothed spectra produced by FastSpeech 2.

At a high level, I don't understand why we would want to solve the problems with FastSpeech in this particular way, and the stated goals of being faster than other diffusion methods do not seem to necessarily have been achieved given the empirical results.

---

### Official Review · Reviewer_Zr8k · 2022-10-26

**Confidence:** 4
**Clarity, Quality, Novelty And Reproducibility:** 1. This paper studies to improve the …
**Correctness:** 4
**Technical Novelty And Significance:** 2
**Empirical Novelty And Significance:** 2
**Recommendation:** 5

**Strength And Weaknesses:**

Strengths
1. The method generates the residual between the output of an existing TTS model and the groundtruth, which makes it efficient compared to those generating speech from scratch.

2. The method is flexible to apply to other existing TTS models without re-training them.

3. Experiments show better MOS scores with the similar RFT and better RFT at similar MOS scores.

Weakness
1. Section 3.1 says that  “The generated residual contains high-frequency details and avoid over-smoothness in the estimated mel-spectrogram, and thus can improve the sample quality.”. When first reading this, I feel It’s unclear what design in ResGrad makes it produce high-frequency details. There is a lack of context before making this claim. I think the reason is because of the way to calculate the residual in training, which is introduced in the later Section 4.1.

2. The paragraph “Calculation of Residual” in Section 4.1 explains why using  ground-truth or predicted pitch/duration to calculate the residual. I think the explanations can be improved. The reason for using groundtruth pitch in calculating residuals is to make ResGrad not learn to correct pitch and reduce its learning burden. However, in inference, the ResGrad still conditions on the mel-spectrogram generated by the predicted pitch. The mel-spectrogram generated with the predicted pitch may be different from the one generated with groundtruth pitch.  If ResGrad is trained to improve the later one, how can we make sure it performs well on the former one in inference?

3. The first paragraph of Section 5.2 mentions “As it can be seen, when we use pitchPred, the residual related to pitch information is not generated as it is in the ground-truth residual samples.” I feel this contradicts the statement in Section 4.1 that “However, if we calculate the residual with predicted pitch as shown in Figure 2(b), ResGrad will simultaneously correct the pitch in estimated samples towards the ground-truth samples”. My understanding is that if using the pitchPred, ResGrad will learn to generate pitch related information to correct the pitch. Table 2 also seems to contradict the descript “Thus, we use ground-truth pitch to calculate the residual.” in Section 4.1. Table 2 shows using pitchGT gives worse results.

4. Regarding Figure 3, are the two residual examples (row 2) generated by the same ResGrad trained with the pitchGT? Or are they trained with pitchGT and pitchPred, respectively?

5. What’s the relation of rows 1 and 2 in Figure 4? Are row 1 one example and row 2 another example? What do the red rectangles mean?


**Summary Of The Paper:**

This work proposes ResGrad to improve the existing TTS models. Specifically, it uses a DFM to learn to generate the residual between an existing TTS model’s output and the groundtruth spectrogram. The residual is then added back to the TTS model’s output to get the refined output. Experiments on three datasets (LJSpeech, LibriTTS, and VCTK) show high sample quality and inference speed.


**Summary Of The Review:**

It provides a new view in improving the efficiency of DFMs in speech synthesis. The method is simple and results look promising. The paper can be improved if it can refine some technical descriptions and experiment details.

---

### Official Review · Reviewer_TPsB · 2022-10-26

**Confidence:** 4
**Clarity, Quality, Novelty And Reproducibility:** The paper is well-written and present…
**Correctness:** 3
**Technical Novelty And Significance:** 2
**Empirical Novelty And Significance:** 2
**Recommendation:** 5

**Strength And Weaknesses:**

Strength:
* Results shown on a bunch of datasets
* Good practical application

Weakness:
* Limited Novelty
* Poor baseline results

**Summary Of The Paper:**

Authors propose diffusion-based refinement of an existing TTS system. Two competing approaches are non-iterative TTS (i.e. approaches which don’t use diffusion process) and diffusion based models.

Authors show improvements in speech quality compared to DDPM based models at similar inference speed.

It is worth noting that the suggested approach is not a new method to speed up diffusion process, rather a TTS system with fast sampling and high quality synthesis.

**Summary Of The Review:**

The paper presents a simple idea of starting the diffusion process to refine the predictions of an existing TTS network. While the authors show that the idea indeed leads to improvement in the inference speed, there are very limited things to learn from the paper. Further study related to use of different residuals e.g. more than one base TTS model -- would strengthen the paper.

I am curious about the baseline MOS reported by the authors -- the variance in the reported MOS is considerably higher compared to other research papers.

---

### Decision · Program_Chairs · 2023-01-20

**Decision:**

Reject

**Justification For Why Not Higher Score:**

All four reviews recommend rejection, with the highest quality reviews making the strongest arguments.

I found the points about the model being poorly motivated and the architecture limiting the usefulness of the DDPM to be quite persuasive.


**Justification For Why Not Lower Score:**

N/A

**Metareview: Summary, Strengths And Weaknesses:**

# Summary

This paper proposes using a denoising diffusion probabilistic model to refine the output of an existing TTS model, FastSpeech2. The rationale for this architecture is that the refinement task is simpler than a full synthesis task, and this reduced complexity means faster execution. Tests on the single-speaker LJSpeech corpus and the multi-speaker LibriTTS and VCTK corpora show that the proposed ResGrad method achieves a better tradeoff between sample quality and inference time compared to other methods for accelerating DDPMs.

# Strengths

- Simple method for using a DDPM to improve the quality of FastSpeech2.
- Paper is clearly written and easy to understand.

# Weaknesses

- Model is not well-motivated: one of the main advantages of DDPMs is their ability to generate multimodal distributions, but in the case of TTS one of the main sources of information to select between modes is the input text, and in the proposed configuration the DDPM does not have access to text.
- The proposed model is quite specific to FastSpeech2 and may not be readily adaptable to other models.
- The proposed model is very similar to GradTTS, and does not appear to perform better. To tell if ResGrad outperforms GradTTS, it would have been necessary to perform CMOS experiments, and this was not done.


**Summary Of Ac-Reviewer Meeting:**

N/A